# Evolution of the Antibiotic Resistance Levels, Multi-Resistance Patterns, and Presence of Antibiotic Resistance Genes in *E. coli* Isolates from the Feces of Breeding Hens during the Rearing Period

**DOI:** 10.3390/antibiotics13080753

**Published:** 2024-08-10

**Authors:** Alejandro Fenollar-Penadés, Pablo Catalá-Gregori, Vicente Tallá-Ferrer, María Ángeles Castillo, Miguel García-Ferrús, Ana Jiménez-Belenguer

**Affiliations:** 1Centro Avanzado de Microbiología de Alimentos, Universitat Politècnica de València, C/Camí de Vera s/n, 46022 València, Spain; alejandro.fpenades@gmail.com (A.F.-P.); migarfe1@upv.es (M.G.-F.); anjibe@upvnet.upv.es (A.J.-B.); 2Centro de Calidad Avícola y Alimentación Animal de la Comunidad Valenciana (CECAV), CEU Universities, Universidad CEU Cardenal Herrera, 46115 Alfara del Patriarca, Spain; p.catala@cecav.org; 3Grupo SADA, Ronda de Poniente 9, 28760 Tres Cantos, Spain; vtalla@sadavc.es

**Keywords:** *Escherichia coli*, antibiotic resistance, third-generation cephalosporin resistance, poultry, antibiotic resistance genes, ESBL, PMQR, breeding hens

## Abstract

The food chain acts as an entry point for antibiotic resistance to reach humans and environment. Because of the importance of the poultry sector, we investigated the prevalence and evolution of antibiotic resistance in *Escherichia coli* isolates from a series of 14,500 breeding hens and their farm environment during the rearing period. Samples included meconium from one-day-old breeders and fecal samples and boot swabs from the breeding sheds of pullets and adult hens. All *E. coli* isolates from one-day-old chicks, 77% from feces and 61% from boot swabs, were resistant to at least one antibiotic. Cefotaxime and multi-drug resistance in fecal isolates decreased during the rearing period from 41.2% and 80.8% in one-day-old chicks to 3.8% and 33.8% in adults. All genes studied were detected in *E. coli* from feces and boot swabs, the most common being *bla*_TEM_ (75%), *bla*_SHV_ (72%), and *qnr*B (67%). *bla*_CMY-2_ was detected in 100% of one-day-old breeders. The combination of at least one cephalosporin and one quinolone resistance gene was detected in 68.7% of fecal and boot swab isolates. Our results highlight the need to monitor the prevalence of antibiotic resistance on farms and to take appropriate measures to reduce the risk to public and environmental health.

## 1. Introduction

Antimicrobial resistance (AMR) is a rising global health threat. It is estimated that bacterial AMR was directly responsible for 1.27 million global deaths in 2019 and contributed to 4.95 million deaths. In addition, the World Bank estimates that AMR could result in USD 1 trillion to USD 3.4 trillion gross domestic product (GDP) losses per year by 2030 [1].

The survey and control of AMR from a One Health perspective are essential for managing this public health emergency [2]. The food chain acts as a main entry point for AMR to reach humans and re-circulate in the ecosystem. Resistant bacteria can be transferred through the whole food chain, from primary production to consumers, via food, the environment, and direct contact with animals [3]. Bacteria can also act as a source of resistance genes that can be transferred to other bacteria, including human and animal pathogens [4]. Studies around the world show an increase in the isolation of AMR bacterial strains from food-producing animals and meat [5,6,7]. Specially, resistance against medically important antibiotics, such as cephalosporins and quinolones, is critical [8]. The mandatory surveillance of antimicrobial-resistant bacteria in food animals and food in the European Union (EU) [5] demonstrates the importance of these bacteria as a public health threat.

*Escherichia coli* (*E. coli*) is a natural inhabitant of the gut in poultry and one of the main microorganisms responsible for the spread of resistant genes among the environment, animals, and humans [9,10]. *E. coli* has been proposed as the main indicator of the presence of antibiotic-resistant bacteria and clinically relevant resistance genes [4,11], due to its presence in animal and human guts, clinical relevance, and ability to acquire conjugative plasmids. Moreover, commensal strains of *E. coli* reflect population exposure to antimicrobial selection pressure and thus can provide continuous evidence of trends [12]. *E. coli* is especially useful to monitor extended-spectrum β-lactamase (ESBL)-producing bacteria, because ESBL-producing *Enterobacteraceae* are the major cause of resistance to expanded-spectrum β-lactam antibiotics [4,13].

Beta-lactams account for 60% of all antibiotics used worldwide and are one of the most widely prescribed antibiotic classes [14]. The most frequent form of resistance to β-lactam antibiotics is the production of β-lactamases, mainly extended-spectrum β-lactamases (ESBLs) and plasmidic AmpC (pAmpC) β-lactamases, which can hydrolyze penicillin and cephalosporins [15]. SHV and TEM are among the most frequently found ESBL gene types, and their presence in plasmids and other mobile genetic elements facilitates their dissemination in all environments [16]. Concerning pAmpC, the most common β-lactamases are the CMY-2 type, which, in contrast to ESBLs, are less affected by β-lactamase inhibitors [17].

Th eintensive use of β-lactam antibiotics in human and veterinary medicine has led to the spread of extended spectrum β-lactamase (ESBL)-producing resistant bacteria. The World Health Organization (WHO) has indicated that ESBL-producing *Enterobacteriaceae* are among the world’s most serious and critical threats in the 21st century [18]. Regarding this threat, the European Food Safety Authority (EFSA) considers the presence of beta-lactamase, especially ESBL-producing *E. coli* in poultry, a public health hazard [19].

On the other hand, resistance to quinolones is usually due to a chromosomal mutation leading to reduced target susceptibility. However, some plasmids can carry genes that confer resistance to these antibiotics, known as plasmid-mediated quinolone resistance (PMQR) [20].

Third-generation cephalosporins and fluoroquinolones are categorized as highest-priority critically important antimicrobials (hpCIAs), because they are commonly used in humans [8]. In 2019, 54% of *E. coli* strains isolated from humans in Europe was resistant to at least one family of antibiotics, mainly fluoroquinolones and beta-lactams [21]. In recent years, resistance to cephalosporins and quinolones has been detected in *E. coli* isolated from food-producing animals, including poultry [22,23,24]. Moreover, the association between ESBL and PMQR genes is frequent in resistant bacteria [25]. This justifies the monitoring of combined resistance to these classes of antimicrobials in food-producing animals [4].

Animal food products play an important role in the human diet, and their demand and production are rising worldwide. Poultry meat is one of the most consumed meat products. In 2020, poultry meat production represented almost 40% of global meat production [26] and its consumption is projected to increase by 17.8% by 2030, according to the OECD-FAO [27].

There is strong evidence that the use of antibiotics in poultry production has led to the development of high resistance levels in the microbiome of animals, increasing the risk of transfer of these resistances from poultry-associated bacteria to human pathogens, with potentially serious consequences for public health [28]. Moreover, poultry farms have been shown to be a source of resistance spread to the environment [29]. Despite the restrictions on antimicrobial use in many countries, there have been multiple reports of antibiotic-resistant bacteria associated with poultry, which present food safety concerns [30]. At present, there is a growing interest in understanding the evolution of antibiotic resistance, not only in broilers, but throughout the entire production pyramid [31,32,33,34], to implement preventive measures at an early stage. Thus, in this study, we aimed to determine the prevalence of antibiotic resistance in *E. coli* isolates obtained from a commercial-breeder hen farm throughout different stages of production (from one-day-old breeders to adult hens), as well as to detect cephalosporin and quinolone resistance genes in those *E. coli* isolates that showed resistance to cephalosporins and/or ciprofloxacin.

## 2. Results 

### 2.1. Antibiotic Resistance Prevalence 

A total of 383 isolates was identified as *E. coli* by the API 20E system (ID > 99%) and subsequently investigated for antibiotic resistance: 68 from one-day-old breeders; 90 from pullets’ fecal samples, 80 from boot swabs from pullet breeding sheds, 80 from adult hens’ fecal samples, and, finally, 65 isolates from adult hens’ shed boot swabs. Table 1 and Figure 1 show the results of the antibiotic resistance levels.

#### 2.1.1. Antibiotic Resistance Prevalence in *E. coli* Isolated from One-Day-Old Breeders

In the isolates from one-day-old breeders, extremely high levels of resistance were observed for ampicillin (AMP, 100% of isolates) and tetracycline (TE, 98.5%) and very high (>50%) for both aminoglycosides, gentamycin (CN) and streptomycin (S). No isolate showing resistance to quinolones (nalidixic acid, NA, and ciprofloxacin, CIP) and chloramphenicol (C) was observed. Levels of resistance to third-generation cephalosporin (3GC) among the isolates were 41.2% for cefotaxime (CTX) and 10.3% for ceftazidime (CAZ). In addition, the percentage of isolates with intermediate susceptibility to both antibiotics was 69.1% for CAZ and 51.5% for CTX. All isolates resistant to CTX and/or CAZ tested positive for the ESBL phenotype. 

#### 2.1.2. Antibiotic Resistance Prevalence in *E. coli* Isolated from Fecal Samples of Pullets and Adult Hens

In *E. coli* isolates from the fecal samples of pullets and adult hens, extremely high resistance levels were observed for AMP (76.9%) and TE (63.1%), followed by high resistance levels for S (22.9%) and NA (10.6%). For the remaining tested antibiotics, low resistance levels (<2%) were observed in all cases. Again, all resistant isolates to both CTX and CAZ tested positive for the ESBL phenotype. 

By age groups, in *E. coli* isolated from pullets, extremely high levels of resistance were observed for AMP (80%) and TE (70%), while low levels were detected for NA (10%), S (7.8%) and CN (2.2%). For the rest of the antibiotics, no resistant isolates were obtained. In the case of the isolates from adult hens, resistance was extremely high for AMP (73.8%), very high for TE (56.3%), high for CN (40%), and moderate for NA (11.3%). For the rest of the antibiotics, the resistance level was low (below 10%). All the isolates were susceptible to CN.

When statistical analysis was performed, isolates from one-day-old hens showed higher levels of resistance to five antimicrobials than those from pullets and adults: CTX (χ^2^ = 67.123, *p* = 0.000), CAZ (χ^2^ = 11.825, *p* = 0.0027), CN (χ^2^ = 107.643, *p* = 0.0000), S (χ^2^ = 60.017, *p* = 0.0000), and TE (χ^2^ = 29.505, *p* = 0.0000). Significant lower levels were found for AMP (χ^2^ = 19.865, *p* = 0.000) and NA (χ^2^ = 7.884, *p* = 0.0194), which were higher in pullets and adults, respectively.

#### 2.1.3. Antibiotic Resistance Prevalence in *E. coli* Isolated from Boot Swab Samples 

For total boot swab isolates, the antibiotics that presented the highest resistance levels were AMP (60.9%) and TE (33.1%). A high resistance level to NA was also observed (25.6%). For cephalosporins, the resistance was higher for CTX (6.4%) than for CAZ (1.4%). As in fecal samples, all isolates resistant to CTX and/or CAZ presented the ESBL phenotype.

The resistance rates detected in the *E. coli* from pullets and adults for AMP, S, and TE were significantly higher for fecal samples than for boot swabs (χ^2^ = 10.686, *p* = 0.0011; χ^2^ = 9.345, *p* = 0.0022, and χ^2^ = 38.012, *p* = 0.0000, respectively). Conversely, for CTX, NA, and C, resistance levels were significantly higher in boot swab isolates (χ^2^ = 4.214, *p* = 0.0401; χ^2^ = 11.171, *p* = 0.0008, and χ^2^ = 7.315, *p* = 0.0068, respectively).

### 2.2. Resistance Profiles and Multi-Resistance (MDR) Patterns

The resistance patterns of isolates exhibiting at least one resistance are shown in Table 2.

Among isolates from one-day-old breeders, 12 different patterns were obtained, the most frequent being AMP/CN/S/TE (30.9%), AMP/TE (19.1%), and AMP/CTX/CN/S/TE (11.8%). Among isolates from pullet fecal samples, 8 profiles were observed, mainly AMP/TE (28.2%), AMP (22.2%), and TE (14.4%), whilst in adult hen isolates, 16 profiles were seen, AMP/S/TE (28,8%) and AMP/TE (16.3%) being the most frequently detected. For boot swab isolates, 22 profiles were identified. Among isolates from pullets, the most prevalent profiles were AMP (27.5%) and NA (11.3%); for isolates from adult hens, the most common profiles were AMP/NA/TE (15.4%), AMP/NA (13.9%), and AMP/S/TE (12.3%).

Regarding the prevalence of multi-resistance (MDR) (Figure 2), a total of 80.8% of *E. coli* isolated from box bottoms (one-day-old chickens), 13.5% from pullets, and 35.1% from adult hens were multi-resistant. Nineteen MDR different profiles were observed, 11 of them in box bottom isolates, AMP/CN/S/TE being the most frequent (21 isolates, 30.9%). Five MDR profiles were detected in isolates from fecal samples, the most common profile being AMP/S/TE, present in 7 pullet and 23 adult isolates (15.5% and 32.5%, respectively). Finally, 23.4% of boot swab isolates were MDR, with 9 different profiles detected. MDR levels were lower in pullets than in adults (11.3% and 38.5%, respectively).

### 2.3. Prevalence and Profiles of Antibiotic Resistance Genes (ARGs)

DNA from 132 isolates showing resistance to cephalosporins and/or quinolones was analyzed by PCR for the detection of β-lactam resistance and PMQR genes. This included 68 *E. coli* isolates from one-day-old breeders, 41 from feces, and 23 from boot swabs (Table 3).

In all *E. coli* isolates from one-day-old breeders, only the *bla*_CMY-2_ gene was observed. In the isolates from feces, all the studied genes were detected. The most prevalent was *bla*_TEM_ (90.2%) followed by *bla*_SHV_ (87.8%), *qnr*B (85.4%), and *qnr*S (24.4%). The *bla*_CMY-2_ gene was observed at a 2.4% rate. In boot swab isolates, only 8.7% was negative for all the genes tested. The genes with the highest rates, as in feces, were *bla*_TEM_ (47.8%) and *bla*_SHV_ (43.5%). The *bla*_CMY-2_ gene was positive in 21.7% of the isolates. Figure 3 shows an example of the detection of β-lactam resistance genes in *E. coli* isolates of different origins and positive *E. coli* controls. 

For quinolone resistance genes, the prevalence of both genes was similar in boot swab isolates (39.1% for *qnr*S and 34.8% for *qnr*B), while, in feces, the prevalence of *qnr*B was significantly higher than that of *qnr*S (χ^2^ = 84.926, *p* = 0.0000).

Regarding the number of resistance genes observed per isolate (Table 4), only one gene was detected in all the one-day-old breeders isolates. In fecal samples, most isolates carried three genes (82.9%), followed by two (12.2%) and four genes (4.9%). In boot swabs, the presence of one gene was the most frequent (43.5%), followed by three (26.1%), and four or five genes (4.3%). 

Concerning the ARG profiles, 10 different profiles were observed in fecal isolates, with ‘*bla*_SHV_-*bla*_TEM_-*qnr*B’ being the most common, present in 70.7% of isolates. In boot swabs isolates, nine different profiles were identified, ‘*bla*_SHV_-*bla*_TEM_-*qnr*B’ also being the most frequently detected (26.1%), followed by ‘*bla*_CMY-2_′ and ‘*qnr*S’ (both at 17.4%). 

## 3. Discussion

In this work, we investigated the levels of antimicrobial resistance in *E. coli* isolates from a commercial broiler farm at different stages of production. Monitoring AMR in commensal *E. coli* isolated from food-producing animals provides information on the reservoirs of resistant bacteria that could potentially be transferred to humans and also provides indirect information on the presence of resistance genes that could be transferred to bacterial pathogens. It is therefore relevant to both public and animal health [19].

When one-day old chickens fecal samples were analyzed, we found extremely or very high levels of resistance to most of the ‘highest priority critically important’ or ‘critically important’ antimicrobials analyzed in this work [8]. Moreover, high levels of resistance or intermediate susceptibility to third-generation cephalosporin was detected among the isolates. Resistance against these antibiotics has also been observed in bacteria isolated from few-days-old chicks by other authors, what suggests that early-stage hens can be a source of antibiotic resistance in the farm environment. In a similar study, Moreno et al. (2019) [31] reported considerable AMR rates, but they were overall lower than those obtained in the present study, and no resistance to CTX or CAZ was observed, although they did detect resistance to CIP and NA. In a previous study on broilers [32], we observed high levels of antibiotic resistance in *E. coli* isolated from the meconium of newly hatched chicks, the main resistances rates being against NA (80%), AMP (70%), and TE (30%). In addition, 16.7% of isolates was CIP-resistant. However, unlike the present study, no cephalosporin-resistant *E. coli* isolate was observed.

The presence of AMR in one-day-old hens at a breeding farm is a cause for concern. Several studies [33,34] have shown the transmission of AMR throughout the hen production system, which may be due to two factors: (a) a possible vertical transmission from parents to the offspring, caused by a possible infection in the hen’s uterus during egg formation, or by fecal contamination in the cloaca during egg-laying [35]; and (b) transmission at the hatcheries. Zurfluh et al. (2014) [33] probed the presence of genetically similar plasmids in ESBL-producing *E. coli* isolated from different points of the production system. In another study in Sweden [34], the presence of an *E. coli* clone in all the levels of the pyramid production system was also detected, supporting the hypothesis that transmissions may occur in the hatcheries due to bacteria resistant to cleaning products, which are co-selected and transmitted to successive hatchlings and adults from one animal to another or through environment. Our methodological approach does not allow us to identify the specific pathways through which AMR *E. coli* reached and spread in the studied farm. More specific studies, tracking different clonal *E. coli* populations over time, should be undertaken to achieve this goal.

Because of their importance in human medicine [18], the high observed rates of resistance and intermediate susceptibility to 3GC we found in one-day-old hens are remarkable. Similar to us, other authors also detected high resistance levels to these antibiotics in *E. coli* isolated from both one-day-old and one-week-old chicks, thus suggesting that 3GC resistance is present in hens in their first moments of life [31,32,35]. However, most of these studies usually looked for cephalosporin-resistant strains using selective enrichment (generally with CTX) as a previous step to bacteria isolation. Thus, their results are not comparable to the resistance rates obtained in ours, in which no selective step was used. Nonetheless, it shows that 3GC resistance is more common than expected, given that their use is not extended to the veterinary field, and suggests the possibility of the off-label use of cephalosporin in hatcheries [36]. The EFSA indicated that there were suspicions of off-label ceftiofur use in one-day-old chicks at the hatcheries with a prophylactic purpose, even though the use of cephalosporin in poultry is not authorized in the EU at present [37]. Thus, it is important to take the necessary hygienic measures at the hatcheries to reduce, as much as possible, the rates of resistance transfer and avoid using practices that may favor the selection of resistant strains, such as the unnecessary use of antibiotics [38].

In *E. coli* isolates from fecal samples in pullets and adults, the overall resistance levels were not high, except for ampicillin, tetracycline, sulfamethoxazole, and nalidixic acid. Resistance to cephalosporins was sporadic. This result is in accordance with that obtained by other authors [33,36] and demonstrates that, even though it is not commonly observed, cephalosporin resistance is present in poultry. Furthermore, since resistant *E. coli* can remain on the farm, there may be a risk of cross-contamination between incoming and outgoing flocks [39].

Comparing our AMR prevalence data with the antimicrobial resistance in commensal *E. coli* in poultry reported by the EFSA and ECDC [4], we obtained different AR levels. In Europe, most AMR is observed for quinolones (46.7% for CIP and 43.6% for NA), followed by AMP (41.8%) and TE (30.5%). The resistance levels for C (9.6%), CN (5.1%), CTX (1.1%), and CAZ (1.1%) are low. In Spain, however, the reported results are generally in the European average, quinolones being the antibiotics with the highest level of resistance (62.9% for CIP and 54.7% for NA), followed by high levels for AMP (31.8%) and TE (30%), moderate levels for CN (7.6%) and C (8.2%), and low levels for CTX (0.6%) and CAZ (0.6%). The main differences between our data and those reported by the EFSA are observed in the quinolone resistance percentages. These results are not surprising. The EFSA report highlights that resistance levels greatly differed among reporting countries and antimicrobials, showing important variations between and within food-producing animal populations and countries [4], due mainly to the absence of harmonized approaches using the same methodology among countries for survey studies [40]. It should also be considered that our study focused on breeding hens while the EFSA reported data obtained from broilers and that the rearing process can differ among farms, thus affecting AMR spread and levels [41,42].

In a work carried out in Spain [32] among laying hens, resistance levels similar to those obtained in our study were observed, except for TE and AMP, for which the resistance rates were lower. For pullets, the authors reported 21% resistant isolates for TE and below 20% for AMP, CIP, and NA; under 10% for C; less than 5% for CN; and sporadic resistance for CTX and CAZ (below 3% for both). In a study from Switzerland conducted on different laying-hen farm systems [43], no resistance to cephalosporins was observed, and the resistance rates observed were inferior to those observed in this work, except for quinolones, for which 16.7% of the isolates was resistant. In another study from Norway, concerning quinolone resistance [44], low levels were observed in *E. coli* isolated from feces and boot swabs from broilers (3.6%) and laying hens (0.5%).

In this work, all the isolates that were resistant to CTX and CAZ showed the ESBL phenotype. Our results are in accordance with other studies that detected isolates with this resistance profile in broiler farms [45,46] and in parent flocks of chickens [47], where ESBL/AmpC-producing *E. coli* was observed at all points of the pyramid production system both with and without a previous selective enrichment step with cefotaxime. All these results demonstrate that cephalosporin-resistant *E. coli* can be found at all stages in the farm production system, which is a public health concern, given the clinical importance of these antibiotics and the likelihood of their spread to consumers through the food chain [29]. Fortunately, the prevalence of resistance to cephalosporins in *E. coli* from adult poultry populations is generally low, and surveillance studies from different European countries have reported a decrease in these resistances [4,42]. However, it is important to keep undertaking measures focused on reducing antibiotic resistance, especially cephalosporin, in primary production.

Comparing resistance occurrence in the isolates from one-day-old breeders and from fecal samples from pullets and adult hens, the statistical analysis showed a clear dependence on the age of the hens, confirming a reduction in the resistance levels with increasing age for all antimicrobials, except for AMP and NA, which were higher in pullets and adults, respectively. As indicated in Section 4, only two antibiotic treatments were administered to the hens during our study. This restrictive protocol aims to prevent antibiotic pressure in the farm environment. The lack of selective pressure avoids AMR spreading among the bacteria population. Moreover, under normal conditions, changes in the intestinal microbiota occur during the growth of the hens, from more transient populations in early stages of their development to more mature bacterial populations as they grow [48]. These changes in the bacterial population and the absence of antibiotic pressure could explain the AR reduction in one-day-old to adult hens.

By studying *E. coli* isolated from boot swab samples, we can estimate the resistance levels among the bacteria population in the farm environment. Although the antibiotics that presented the highest resistance levels were the same (AMP and TE), in this case, a high resistance level to NA was observed. Regarding the rest, the level of resistance was low or moderate. 

The resistance levels in the environment for most of the antibiotics, including CTX and CAZ, remained stable during the hens’ growth, showing no statistical dependence with the age of the animals. Moreover, for CTX, NA, and C, resistance levels were significantly higher in boot swab isolates than in fecal samples. Given the low use of antibiotics during rearing, this suggests that certain resistance determinants can remain in the environment, even in the absence of selective pressure, and spread through the *E. coli* population of the farm in case of favorable events, such as co-selection mediated by antibiotic treatments or exposure to disinfectants [49]. Dame-Korevaar et al. (2017) [50] detected cephalosporin resistance in all the farm environmental samples they analyzed. They also reported that, before the introduction of the hens, the laying area was negative for cephalosporin resistance, suggesting the hens as likely having introduced the resistant strains.

A decrease in resistances only to AMP or NA and increased combined profiles and multi-resistance levels were observed in *E. coli* isolates from boot swabs in adults, what suggests horizontal gene transfer events along the pyramid production system, leading to the combination of different resistances and favoring the spreading of multiple resistances among bacterial populations [51,52].

In all *E. coli* isolates from one-day-old breeders, only the *bla*_CMY-2_ gene was observed. Other authors have reported that the increased presence of this gene in chicks occurs at an early stage of their life [36,53]. The reduction observed in *bla*_CMY-2_ prevalence in isolates from pullets and adults from that observed in one-day-old breeders suggests that, despite its presence during the hens’ first moments of life, in conditions of a low use of antibiotics, this gene did not persist in the *E. coli* populations from the hens in the farm environment. Another possible reason is the use of antibiotics at early stages and the reduction in use at later stages, which allow the susceptible strains to persist. Some authors have also reported similar data. Apostolakos et al. (2019) [53] found a high prevalence of *bla*_CMY-2_ genes in one-day-old chicks and a drop in the prevalence in the laying phase. In a study on parental hens, we also observed a decrease in *bla*_CMY-2_ gene prevalence, from 91% in fecal isolates of one-week-old chicks to 1% in adult hens, at the end of the study [32]. According to this result, the high prevalence during the first week of life could be due to the vertical transmission from the grandparent flock or contamination from other specific sources, such as the hatchery and transport. 

In the *E. coli* isolates from feces and boot swabs, all the studied genes were detected, the most prevalent being *bla*_TEM_ and *bla*_SHV_. Many other studies are in accordance with these results [36,42]. Manageiro et al. (2017) [54] found TEM and SHV family genes in around 33% and 46%, respectively, of these isolates that were non-susceptible to cephalosporin. Blaak et al. (2015) [55] detected *bla*_TEM_ and *bla*_SHV_ genes in approximately 30% of *E. coli* with the ESBL phenotype. 

For quinolone PMQR resistance genes (*qnr*B and *qnr*S), the prevalence of both genes was similar (39.1% for *qnr*S and 34.8% for *qnr*B), whereas in feces, the prevalence of *qnr*B was significantly higher than *qnr*S (χ^2^ = 84.926, *p* = 0.0000). These results differ from other authors’: Jones-Dias et al. (2013) [56] observed less than a 3% prevalence of PMQR genes in *E. coli* and *Salmonella* isolated from poultry and pigs in Portugal. In a longitudinal study conducted in Sweden, PMQR genes were not observed in NA-resistant *E. coli* [57]. In Italy, a prevalence of PMQR genes in *E. coli* strains of 5% in broilers and 4% in laying hens was reported [58]. The differences observed between the present study and others could be due to variations in the microbiota of the animals, which can be affected by environmental differences, such as feed, antibiotic usage, and the hatchery of origin [48].

The presence of ARGs in isolates from the farm environment may pose a risk because of their transmission to other bacteria, including human and animal pathogens [59,60]. Merchant et al. (2012) [61] reported the survival of *E. coli* in the soil even 7 months after fertilization with broiler manure, which confirms the risk of spreading antibiotic resistance to the environment if proper measures are not taken. 

Most of the isolates from feces shared the same ARG pattern, while the rest had a low occurrence rate. On the contrary, in boot swabs, there was no profile with such a high rate as the dominant one in feces. This indicates that, in practice, environmental isolates present higher heterogenicity than fecal isolates, possibly because their contact with strains in the environment facilitates genetic transfer and, thus, a higher diversity [58].

It is remarkable that most of the *E. coli* tested carried at least one of the three ESBL/AmpC genes analyzed, and at least one of the two PMQR genes. This was especially noticed in fecal isolates, where all of them presented a combination of ESBL and PMQR genes. Other authors have also reported the association between ESBL and PMQR genes observed in the present study [62,63]. These findings suggest the possibility of a co-existence in the same genetic element of *qnr* and ESBL or AmpC genes [25].

Overall, our study found high levels of resistance and ARG carriages among the tested isolates. Multi-resistant *E. coli* was found at every stage of hen rearing, and ARGs were detected in all 3GC-resistant isolates. Remarkably, a decreasing trend in these three parameters was observed throughout the hens’ rearing period. However, some limitations of the study must be considered. Although the sampled farm represents typical conditions in our poultry sector, resistance levels may vary among farms due to different rearing methodologies or environmental conditions (e.g., feed, water quality, and housing conditions) that could influence the results [40,41,48]. Moreover, this study focuses on specific ARGs, potentially missing other relevant resistance genes [64]. 

## 4. Materials and Methods

### 4.1. Sampling and Sample Preparation

A batch of 14,500 breeding hens from a commercial breeding farm located in eastern Spain was sampled. The hens were one-day old at the beginning of the experiment and 28 weeks old at the end. During the study, two antibiotics were administered by veterinarians with therapeutic purposes: tylosin was administered at week 4 for 5 days, and amoxicillin was used at week 25 for 4 days. No more antibiotics were used throughout the study.

The batch was sampled five times: on arrival at the farm (S1, one-day-old breeders) and at 4 (S2), 19 (S3), 25 (S4), and 28 (S5) weeks of age, before being transferred to the production farm. Animals from S2 and S3 samplings were considered ‘pullets’, while those from S4 and S5 were named as ‘adult’ hens.

One-day-old breeders’ samples (S1) consisted of 25 g of transport box bottoms containing meconium droppings from one-day-old animals. For the other 4 samplings (S2 to S5), two types of samples were analyzed: the soil residues adhered to a pair of boot swabs and 25 g of composite samples of fecal material taken from the belt collector that removed the manure. Fecal samples were taken in duplicate, refrigerated, and processed within 24 h.

Samples were diluted in peptone water (Buffered Peptone Water (ISO), Scharlau, Spain) and homogenized in a stomacher machine (BagMixer, Interscience, France). In S1, 25 g of box bottoms was mixed with 225 mL of peptone water. The boot swabs were weighed, introduced to 1:10 *w*/*v* peptone water, and mixed. From each composite fecal sample, 25 g was taken and homogenized in 225 mL of peptone water. 

### 4.2. Escherichia coli Isolation

Serial decimal dilutions were made and plated onto Microinstant Chromogenic Coliforms Agar (Scharlau, Barcelona, Spain), which is a selective and differential culture medium for *E. coli*. The plates were incubated for 24 h at 37 °C. Typical *E. coli* colonies (dark blue to violet) were randomly selected after incubation. Seventy colonies, for S1, and 30–40 for the rest of the samples, were recovered. The selected colonies were plated onto non-selective agar media (Plate Count Agar (PCA), Scharlau, Barcelona, Spain) and incubated for 24 h at 37 °C. The isolates were identified as *E. coli* using the API phenotypic identification system (API 20E strips, BioMèriux, Marcy-l’Étoile, France). Confirmed *E. coli* isolates were sub-cultivated in PCA and refrigerated at 4 °C until further analysis. All of them were also frozen at −20 °C in cryovials (Pro-lab Diagnostics Microbank^TM^, Richmond Hill, ON, Canada).

### 4.3. Antimicrobial Susceptibility Testing of E. coli Isolates

Antibiotic susceptibility was tested for all *E. coli* isolates. For this purpose, the disk diffusion technique was used following the Clinical and Laboratory Standards Institute (CLSI) guidelines [65]. Briefly, a 24 h fresh culture of each isolate was collected from plates and resuspended in Mueller–Hilton broth (Scharlab, Barcelona, Spain), adjusting the turbidity to the 0.5 McFarland standard. The suspension was spread onto Mueller–Hinton agar (Scharlab, Barcelona, Spain). Antibiotic disks (Antimicrobial Susceptibility Test Disc, Thermo Fisher Scientific, Waltham, MA, EUA) were placed on the surface and the plates were incubated at 37 °C for 18 h.

The recommendations of the EFSA for the monitoring of antibiotic resistance in commensal *E. coli* isolated from animals [5] were considered for the selection of antibiotics. The antibiotics tested (Antimicrobial Susceptibility Test Discs, OXOID Ltd., England, UK) were ampicillin (AMP, 10 µg), cefotaxime (CTX, 30 µg), ceftazidime (CAZ, 30 µg), ciprofloxacin (CIP, 5 µg), nalidixic acid (NA, 30 µg), chloramphenicol (C, 30 µg), gentamicin (CN, 10 µg), streptomycin (S, 10 µg), and tetracycline (TE, 30 µg).

The extended-spectrum β-lactamase (ESBL) phenotype was checked for colonies resistant to CTX and/or CAZ, according to the CLSI guidelines [66]: resistance to CTX + clavulanic acid (Cefotaxime + Clavulanic acid: CTL 40 µg, Liofilchem Diagnostics, Roseto degli Abruzzi, Italy) and CAZ + clavulanic acid (Ceftazidime + Clavulanic acid: CAL 40 µg, Liofilchem Diagnostics, Roseto degli Abruzzi, Italy) were tested by disk diffusion. If the diameter obtained with the antibiotic/clavulanic acid combination was 5 mm or greater than that without clavulanic acid, the colony was considered ESBL positive.

The prevalence of resistance to each antimicrobial tested was studied and the statistical dependence relationship between resistance, age of the hens, and origin of the samples was assessed. 

The terms used to describe the level of AMR occurrence were the same as those used by the EFSA and the European Centre for Disease Control (ECDC) [4]: rare (<0.1%), very low (0.1–1.0%), low (>1–10.0%), moderate (>10.0–20.0%), high (>20.0–50.0%), very high (>50.0–70.0%), and extremely high (>70.0%).

According to Magiorakos et al. (2012) [66], isolates that showed resistance to 3 or more antibiotics of different classes were considered as multi-resistant (MDR).

### 4.4. DNA Extraction

DNA extraction was performed by thermal lysis. To that purpose, 2–3 colonies from an overnight pure culture of each isolate incubated at 37 °C in PCA media were suspended in 150 µL of TE 1× buffer (TE buffer (1×) pH 7.5, Panreac-AppliChem, Barcelona, Spain) in sterile Eppendorf tubes irradiated with UV for 15 min. The cell suspension was incubated for 10 min at 95 °C in a dry bath. Afterward, the tubes were cooled down with ice for 2 min, centrifuged at 13,000 rpm for 8 min, and the supernatant was transferred to a new sterile and irradiated Eppendorf tube. Quantity (A_260_–A_320_) and quality (A_260_/A_280_ ratio > 1.7) of DNA were assessed by spectrophotometry (Qubit™ 4 Fluorometer, Invitrogen, Walthman, MA, USA) [67]. Extracted DNA was kept at −20 °C until use.

### 4.5. Detection of Antibiotic Resistance Genes (ARGs)

One mPCR for the detection of cephalosporin resistance-related genes *bla*_SHV_, *bla*_TEM_, and *bla*_CMY-2_ was performed according to Colom et al. (2003) [68] and Kozak et al. (2009) [69]. For the detection of PMQR genes *qnr*B and *qnr*S, the mPCR assay described by Cattoir et al. (2007) [70,71] was used (Table 5). As positive controls, we used *Escherichia coli* ATCC 35218 for *bla*_TEM_, *Klebsiella pneumoniae* subsp. *pneumoniae* ATCC 700603 for *bla*_SHV_, *K. pneumoniae* NCTC 13440 for *qnr*S, and two of our own positive *E. coli* strains for *bla*_CMY-2_ and *qnr*B (M1A mec8 and M2C mec6, respectively). MilliQ water was used as a negative control.

### 4.6. Statistical Analysis

Statistical analysis was performed using Statgraphics (Centurion XVII) software (Statpoint Technologies, Inc., Warrenton, VA, USA). Antibiotic resistance and ARG detection were analyzed via a χ^2^ test, using contingency tables to establish any possible dependent correlation between the origin of samples and the age of hens. A probability value of less than 5% was considered statistically significant.

## 5. Conclusions

In summary, in the present study, high rates of multi-resistant *E. coli* were isolated at all phases of hen rearing. However, the prevalence of resistances decreased during the rearing of breeder hens. We observed high rates of cephalosporin resistant *E. coli* isolates from one-day-old breeders without the need for selective culturing, highlighting the suspicion of the off-label use of these antibiotics at the hatcheries. Furthermore, resistance genes against cephalosporin and quinolones were frequently detected in the same isolates, supporting the co-presence of ESBL and PMQR genes in *E. coli* that were non-susceptible to cephalosporins. Both the high rates of multi-resistant isolates and the presence of resistance genes to different antibiotic classes, even in the case of not being exposed to these antibiotic families, highlight the necessity of surveying the prevalence of antibiotic resistance in farm animals and environments, focusing on critically important antibiotics but without neglecting non-critical ones, due to their possible role in co-selection. This approach ensures that proper management and prevention measures can be implemented to reduce antibiotic resistance and its spread.

## Figures and Tables

**Figure 1 antibiotics-13-00753-f001:**
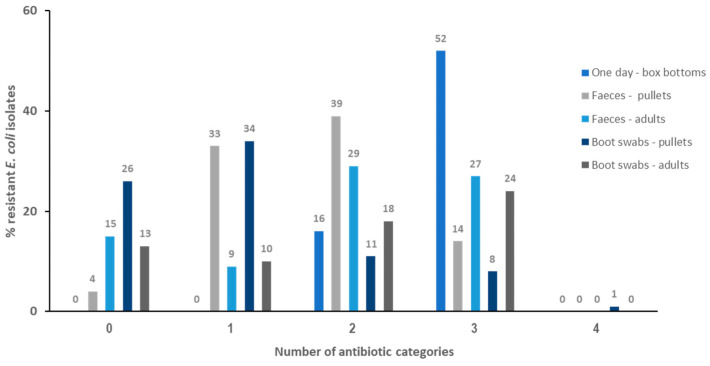
Resistance levels to different antibiotic categories observed among the *E. coli* isolates.

**Figure 2 antibiotics-13-00753-f002:**
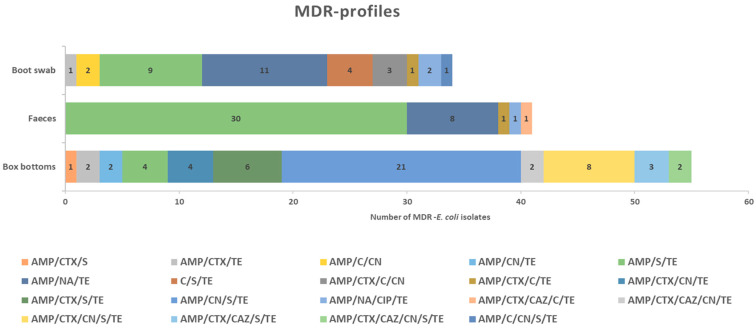
Distribution of MDR profiles in the *E. coli* isolates.

**Figure 3 antibiotics-13-00753-f003:**
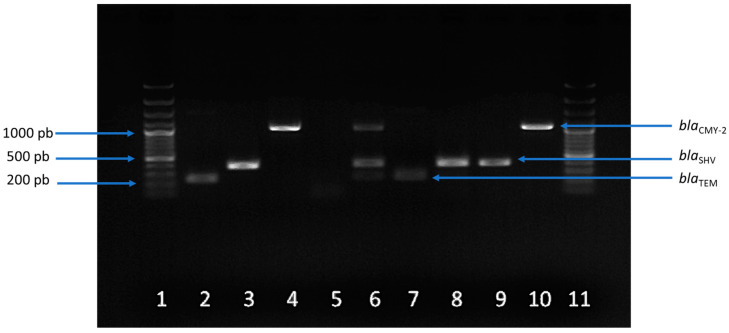
PCR results of β-lactam resistance genes. Lanes 1 and 11: Molecular 100 bp ladder; Lane 2: *bla*_TEM_ positive control; Lane 3: *bla*_SHV_ positive control; Lane 4: *bla*_CMY-*2*_ positive control; Lane 5: Negative control; Lane 6: Sample BS15 (boot swabs, *bla*_TEM_-*bla*_SHV_-*bla*_CMY-2_); Lane 7: Sample F21 (feces, *bla*_TEM_); Lane 8: Sample BS18 (boot swabs, *bla*_TEM_-*bla*_SHV_)*;* Lane 9: Sample F6 (feces, *bla*_SHV);_ Lane 10: Sample BB3 (one-day-old chicken, *bla*_CMY-2_).

**Table 1 antibiotics-13-00753-t001:** Antibiotic resistance levels in the *E. coli* isolates.

Origin	No. of Tested Isolates	Number of Resistant Isolates (%)								
		AMP	CTX	CAZ	CIP	NA	C	CN	S	TE
Box bottoms One-day old	68	68 (100)	28 (41.2)	7 (10.3)	0	0	0	39 (57.4)	45 (66.2)	67(98.5)
Feces										
Pullets	90	72 (80)	0	0	0	9 (10)	0	2 (2.2)	7 (7.8)	63 (70)
Adults	80	59 (73.8)	3 (3.8)	2 (2.5)	2 (2.5)	9 (11.3)	3 (3.8)	0	36 (40)	45 (56.3)
% of resistant isolates		76.9	1.9	1.2	1.2	10.7	1.9	1.1	23.9	63.1
Boot swabs										
Pullets	80	42 (52.5)	4 (5)	1 (1.3)	2 (2.5)	15 (18.8)	6 (7.5)	7 (8.8)	4 (5)	11 (13.8)
Adults	65	45 (69.2)	5 (7.7)	1 (1.5)	1 (1.5)	21(32.3)	6 (9.2)	0	13 (20)	34 (52.3)
% of resistant isolates		60.9	6.4	1.4	2.0	25.6	8.4	4.4	12.5	33.1
TOTAL % of resistant isolates		74.7	10.4	2.9	1.3	14.1	3.9	12.5	27.4	57.4

AMP—ampicillin; CTX—cefotaxime; CAZ—ceftazidime; CIP—ciprofloxacin; NA—nalidixic acid; C—chloramphenicol; CN—gentamicin; S—streptomycin; TE—tetracycline.

**Table 2 antibiotics-13-00753-t002:** Resistance profiles for the *E. coli* isolates.

Profile	No. Isolates (%)				
	Box Bottoms	Feces		Boot Swabs	
	One-Day-Old (*n* = 68)	Pullets (*n* = 90)	Adults (*n* = 80)	Pullets (*n* = 80)	Adults (*n* = 65)
AMP	-	20 (22.2)	4 (5.0)	22 (27.5)	6 (9.2)
NA	-	-	1 (1.3)	9 (11.3)	-
S	-	-	-	1 (1.3)	-
TE	-	13 (14.4)	3 (3.8)	1 (1.3)	2 (3.1)
AMP/C	-	-	1 (1.3)	-	1 (1.5)
AMP/CTX	-	-	-	-	1 (1.5)
AMP/CN	-	2 (2.2)	-	-	-
AMP/NA	-	1 (1.1)	4 (5.0)	3 (3.8)	9 (13.9)
AMP/S	-	-	8 (10.0)	1 (1.3)	-
AMP/TE	13 (19.1)	35 (38.9)	13 (16.3)	5 (6.3)	6 (9.2)
CN/TE	-	-	-	1 (1.3)	-
NA/TE	-	1 (1.1)	1 (1.3)	-	-
S/TE	-	-	1 (1.3)	-	1 (1.5)
AMP/NA/CIP	-	-	1 (1.3)	1 (1.3)	-
AMP/CTX/CAZ	-	-	1 (1.3)	1 (1.3)	1 (1.5)
AMP/CTX/S	1 (1.5)	-	-	-	-
AMP/CTX/TE	2 (2.9)	-	-	-	1 (1.5)
AMP/C/CN	-	-	-	2 (2.5)	-
AMP/CN/TE	2 (2.9)	-	-	-	-
AMP/S/TE	4 (5.9)	7 (7.8)	23 (28.8)	1 (1.3)	8 (12.3)
AMP/NA/TE	-	7 (7.8)	1 (1.3)	1 (1.3)	10 (15.4)
C/S/TE	-	-	-	-	4 (6.2)
AMP/CTX/C/CN	-	-	-	3 (3.8)	-
AMP/CTX/C/TE	-	-	1 (1.3)	-	1 (1.5)
AMP/CTX/CN/TE	4 (5.9)	-	-	-	-
AMP/CTX/S/TE	6 (8.8)	-	-	-	-
AMP/CN/S/TE	21 (30.9)	-	-	-	-
AMP/NA/CIP/TE	-	-	1 (1.3)	1 (1.3)	1 (1.5)
AMP/CTX/CAZ/C/TE	-	-	1 (1.3)	-	-
AMP/CTX/CAZ/CN/TE	2 (2.9)	-	-	-	-
AMP/CTX/CN/S/TE	8 (11.8)	-	-	-	-
AMP/CTX/CAZ/S/TE	3 (4.4)	-	-	-	-
AMP/CTX/CAZ/CN/S/TE	2 (2.9)	-	-	-	-
AMP/C/CN/S/TE	-	-	-	1 (1.3)	-

AMP—ampicillin; CTX—cefotaxime; CAZ—ceftazidime; CIP—ciprofloxacin; NA—nalidixic acid; C—chloramphenicol; CN—gentamicin; S—streptomycin; TE—tetracycline.

**Table 3 antibiotics-13-00753-t003:** Frequency of ARG detection in the *E. coli* isolates.

	Number of Isolates (%)				
Origin	*bla* _TEM_	*bla* _SHV_	*bla* _CMY-2_	*qnr*B	*qnr*S
Box bottoms (*n* = 68)			68 (100)		
Feces (*n* = 41)	37 (90.2)	36 (87.8)	1 (2.4)	35 (85.4)	10 (24.4)
Boot swabs (*n* = 23)	11 (47.8)	10 (43.5)	5 (21.7)	8 (34.8)	9 (39.1)
Total (%)	48 (36.4)	46 (34.8)	74 (56.1)	43 (32.6)	19 (14.4)

**Table 4 antibiotics-13-00753-t004:** ARG profiles present in *E. coli* isolates.

Origin	Profile	Number of Isolates (%)
Box bottoms (*n* = 68)	*bla* _CMY-2_	68 (100)
Feces (*n* = 41)	*bla*_SHV_-*bla*_TEM_-*qnr*B	29 (70.7)
	*bla*_SHV_-*bla*_TEM_-*qnr*B-*qnr*S	2 (4.9)
	*bla*_TEM_-*qnr*B-*qnr*S	2 (4.9)
	*bla*_TEM_-*qnr*S	2 (4.9)
	*bla*_SHV_-*bla*_CMY-2_-*qnr*S	1 (2.4)
	*bla*_SHV_-*bla*_TEM_-*qnr*S	1 (2.4)
	*bla*_SHV_-*qnr*B	1 (2.4)
	*bla_S_*_HV_-*qnr*B-*qnr*S	1 (2.4)
	*bla*_SHV_-*qnr*S	1 (2.4)
	*bla*_TEM_-*qnr*B	1 (2.4)
Boot swabs (*n* = 23)	none	2 (8.7)
	*bla*_SHV_-*bla*_TEM_-*qnr*B	6 (26.1)
	*bla* _CMY-2_	4 (17.4)
	*qnr*S	4 (17.4)
	*bla*_SHV_-*qnr*S	2 (8.7)
	*bla* _TEM_	2 (8.7)
	*bla*_SHV_-*bla*_TEM_-*bla*_CMY-2_-*qnr*B-*qnr*S	1 (4.4)
	*bla*_SHV_-*bla*_TEM_-*qnr*B-*qnr*S	1 (4.4)
	*bla*_TEM_-*qnr*S	1 (4.4)

**Table 5 antibiotics-13-00753-t005:** Primers used for the detection of antimicrobial resistance genes.

Primers	Sequence	Product Size (bp)	Reference
*bla*_SHV_-f	AGGATTGACTGCCTTTTTG	393	[68]
*bla*_SHV_-r	ATTTGCTGATTTCGCTCG		
bla_TEM_-f	TTAACTGGCGAACTACTTAC	247	[69]
*bla*_TEM_-r	GTCTATTTCGTTCATCCATA		
*bla*_CMY-2_-f	GACAGCCTCTTTCTCCACA	1000	
*bla_CMY-2_*-r	TGGACACGAAGGCTACGTA		
*qnr*B-f	GGMATHGAAATTCGCCACTG	264	[70]
*qnr*B-r	TTTGCYGYYCGCCAGTCGAA		
*qnr*S-f	GCAAGTTCATTGAACAGGGT	428	[71]
*qnr*S-r	TCTAAACCGTCGAGTTCGGCG		

## Data Availability

Supporting data for this manuscript are available on request from the corresponding author.

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
