# Peer review of "Evolution of the Antibiotic Resistance Levels, Multi-Resistance Patterns, and Presence of Antibiotic Resistance Genes in E. coli Isolates from the Feces of Breeding Hens during the Rearing Period"

_antibiotics, 2024, doi:10.3390/antibiotics13080753_

Round 1

Reviewer 1 Report

Comments and Suggestions for Authors

Manuscript entitled “Evolution of the antibiotic resistance levels, multi-resistance patterns and presence of antibiotic resistance genes in E. coli strains isolated from the feces of breeding hens during the rearing period” is interesting but require major revisions. Introduction section lack important information about E.coli resistance and some sentences lack flow. Material and method section require some revision in writing; prevalence results need to added, it would be better to include results images/ figures in result and discussion section for better understanding. Overall writing need to be improved before manuscript is accepted

1.      Abstract: Line 23, sentence is confusing, please correct and mention result data here 

2.      Abstract: Please mention results in the form of figures in abstract

3.      Introduction: Add more information about role in E.coli in antibiotic resistance

4.      Line 49-50: why? Give a justification/ add more information

5.      47: Write E.coli full form, followed by abbreviation

6.      Line 52: why focus is on “β-lactam antibiotics” only. Add some background, please check the flow sentences so that they connected to each other.

7.      Line 64: How “resistance to cephalosporins and quinolones” is connected to previous sentances, some sentences are out of place, please make connection

8.      Line 68: Write “Poultry meat”

9.      Line 70: Add 1-2 sentences about antibiotic sentences in poultry and why this research was conducted

1      Line 72-74: These sentences are confusing, please revise

1    Results and discussion: Line :77: 383 E.coli were isolated? How many were identified, please mention API kit result here

       Line 82-86: These sentences belong to methodology section, please mention results of prevalence of E.coli and antibiotic resistance here

1    Line 140: Rephrase the sentence

1     Line 318: If possible include Gel electrophoresis figure here

1     Material and method: Line 374: “were”

1      Line 375: This sentence out of place” A collector belt removed the manure”

          Lin3 395: Please mention names of agars

          Line 400: Why molecular methods were not used to identify E.coli

           Line 405: briefly mention disk diffusion method used

2   Line 432: briefly mention method used

Comments on the Quality of English Language

1.      Line 68: Write “Poultry meat”

1.      Line 72-74: These sentences are confusing, please revise

1.      Line 140: Rephrase the sentence

1.      Material and method: Line 374: “were”

Overall writing need to be improved 

Author Response

REVIEWER 1:

Manuscript has been revised and modified, according to kind comments and suggestions of the reviewer. Changes are yellow marked in the draft:

ABSTRACT:

Line 23, sentence is confusing, please correct and mention result data here. Please mention results in the form of figures in abstract.

The abstract has been modified according to the reviewers' suggestions.

INTRODUCTION:

Add more information about role in E.coli in antibiotic resistance

We have added some more information, supported by references (Lines 48-53):

“Escherichia coli (E. coli) is a natural inhabitant of the gut in poultry and one of the main microorganisms responsible for the spread of resistant genes among the environment, animals, and humans [9-10]. E. coli has been proposed as the main indicator of the presence of antibiotic resistant bacteria and clinically relevant resistance-genes [4,11], due to its presence in animal and human gut, clinical relevance and ability to acquire conjugative plasmids.”

Line 49-50: why? Give a justification/ add more information

More information and another reference have been added to justify this statement (Lines 55-57):

“E. coli is especially useful to monitor extended-spectrum β-lactamases (ESBL) producing bacteria, because ESBL-producing Enterobacteraceae are the major cause of resistance to expanded-spectrum β-lactam antibiotics [4, 13]”.

Line47: Write E.coli full form, followed by abbreviation.

It has been done

Line 52: why focus is on “β-lactam antibiotics” only. Add some background, please check the flow sentences so that they connected to each other

We have added some more background information to clarify this point, and checked the paragraph (Lines 67-72):

“Intensive use of β-lactam antibiotics in human and veterinary medicine has led to the spread of extended spectrum β-lactamase (ESBL) producing resistant bacteria. The World Health Organization (WHO) has indicated that ESBL-producing Enterobacteriaceae, are among the world’s most serious and critical threats of the 21st century [18]. Regarding to this threat, the European Food Safety Authority (EFSA) considers the presence of beta-lactamase, especially ESBL-producing E. coli in poultry, a public health hazard [19].”  

Line 64: How “resistance to cephalosporins and quinolones” is connected to previous sentences, some sentences are out of place, please make connection

We have re-written this paragraph, adding some more information to better connect the sentences (LINES 77-85):

“Third-generation cephalosporins and fluoroquinolones are categorised as highest priority critically important antimicrobials (hpCIAs), because they are commonly used in humans [8]. In 2019, 54% of E. coli strains isolated from humans in Europe were resistant to at least one family of antibiotics, mainly fluoroquinolones and beta-lactams [21]. In recent years, resistance to cephalosporins and quinolones has been detected in E. coli isolated from food-producing animals, including poultry [22-24]. Moreover, the association between ESBL and PMQR genes is frequent in resistant bacteria [25]. This justifies the monitoring of combined resistance to these classes of antimicrobials in food-producing animals [4].

Line 68: Write “Poultry meat”. Done (Line 87)

Line 70: Add 1-2 sentences about antibiotic sentences in poultry and why this research was conducted; Line 72-74: These sentences are confusing, please revise

According to reviewer suggestion, we have added some more information and revise redaction of the final sentence, to better clarify why this research was conducted (Lines 91-105):

“There is strong evidence that the use of antibiotics in poultry production has led to the development of high resistance levels in the microbiome of animals, increasing the risk of transfer of these resistances from poultry-associated bacteria to human pathogens, with potentially serious consequences for public health [28]. Moreover, poultry farms have been shown to be a source of resistance spread to the environment [29]. Despite the restrictions on antimicrobial use in many countries, there have been multiple reports of antibiotic-resistant bacteria associated with poultry, which present food safety concerns [30]. At present, there is a growing interest in knowing the evolution of antibiotic resistance, not only in broilers, but throughout the entire production pyramid [31-34], to implement preventive measures at an early stage. Thus, in this study we aimed to determine the prevalence of antibiotic resistance in E. coli isolates obtained from a commercial breeder hen farm throughout the different stages of production (from day-old breeders to adult hens), as well as to detect cephalosporin and quinolone resistance genes in those E. coli isolates that showed resistance to cephalosporins and/or ciprofloxacin.”

RESULTS AND DISCUSSION:

Line:77: 383 E. coli were isolated? How many were identified, please mention API kit result here

During this work, more than 400 characteristic colonies were isolated and submitted to API 20 E identification. 383 isolates were undoubtfully identified as E. coli by API 20E system and were subsequently analysed for antibiotic resistances.

As requested by reviewer, we have included this sentence in the text (Lines 108-109):

“A total of 383 isolates were identified as E. coli by API 20E system (ID > 99%) and subsequently investigated for antibiotic resistance”

Line 82-86: These sentences belong to methodology section, please mention results of prevalence of E. coli and antibiotic resistance here

According to reviewer’s comment, we have included these sentences in “Material and Methods” Section (Lines 484-492):

“The prevalence of resistance to each antimicrobial tested was studied and the statistical dependence relationship between resistance, age of the hens and origin of the samples was assessed. The terms used to describe the level of AMR occurrence were the same as those used by EFSA and the European Centre for Disease Control (ECDC) [4]: rare (< 0.1%), very low (0.1–1.0%), low (> 1–10.0%), moderate (> 10.0–20.0%), high (> 20.0–50.0%), very high (> 50.0–70.0%) and extremely high (> 70.0%). According to Magiorakos et al. (2012) [66], isolates that showed resistance to 3 or more antibiotics of different classes were considered as multiresistant (MR).”

Line 140: Rephrase the sentence

Sentence has been re-written (Lines 129-131):

“In E. coli isolates from fecal samples of pullets and adult hens, extremely high resistance levels were observed for AMP (76.9%) and TE (63.1%), followed by high resistance levels for S (22.9%) and NA (10.6%).”

Line 318: If possible, include Gel electrophoresis figure here

One gel electrophoresis figure has been included (Line 209)

MATERIAL AND METHOD:

Line 374: “were”. Done (Line 430)

Line 375: This sentence out of place” A collector belt removed the manure”.

It has been re-written (Line 442):

Fecal material taken from the belt collector that removed the manure.”

Line 395: Please mention names of agars

It has been done (Lines 451-452 and 455-456):

“Serial decimal dilutions were made and plated onto Microinstant Chromogenic Coliforms Agar (Scharlau, Barcelona, Spain),”

“The selected colonies were plated onto non-selective agar media (Plate Count Agar (PCA),”

Line 400: Why molecular methods were not used to identify E.coli

Phenotypic characterization by API System is a robust, widely used method recommended by FDA for identifying Enterobacteriaceae (https://www.fda.gov/media/166699/download; https://www.fda.gov/media/177960/download?attachment, etc.). Thus, we decided to use this technique because we find it easy to perform and cost-effective.

Line 405: briefly mention disk diffusion method used

We have added a brief description of the method (Lines 465-470):

“Briefly, a 24 h fresh culture of each isolate was collected from plates and resuspended in Mueller-Hilton broth (Scharlab, Barcelona, Spain), adjusting turbidity to 0,5 McFarland standard. The suspension was spread onto Mueller-Hinton agar (Scharlab, Barcelona, Spain). Antibiotic disks (Antimicrobial Susceptibility Test Disc, Thermo Fisher Scientific, Waltham, MA, EUA) were placed onto the surface and the plates were incubated at 37 °C for 18 h.”

Line 432: briefly mention method used

We have added a brief description of the method (Lines 495-503):

DNA extraction was performed by thermal lysis. To that purpose, 2-3 colonies from an overnight pure culture of each isolate incubated at 37 °C in PCA media were suspended in 150 µL of TE 1x buffer (TE buffer (1x) pH 7.5, Panreac-AppliChem, Barcelona, Spain) in sterile Eppendorf tubes irradiated with UV for 15 minutes. The cell suspension was incubated for 10 minutes at 95 °C in a dry bath. Afterward, the tubes were cooled down with ice for 2 minutes, centrifuged at 13000 rpm for 8 minutes and the supernatant was transferred to a new sterile and irradiated Eppendorf tube. Quantity (A260–A320) and quality (A260/A280 ratio > 1.7) of DNA was assessed by spectrophotometry (Qubit™ 4 Fluorometer, Invitrogen, Walthman, MA, USA) [67]. Extracted DNA was kept at -20 °C until use.”

COMMENTS ON THE QUALITY OF ENGLISH LANGUAGE

Line 68: Write “Poultry meat”. Done (Line 87)

Line 72-74: These sentences are confusing, please revise. They have been revised (see above)

Line 140: Rephrase the sentence. Sentence has been re-written (see above)

Material and method: Line 374: “were”. Done

Overall writing need to be improved

Writing has been carefully checked and improved throughout the text

Reviewer 2 Report

Comments and Suggestions for Authors

The submitted manuscript analyzed different isolates of E. coli from the feces of breeding hens. The text is clear and the results are clearly presented. I would like to highlight the use of statistical tools, which makes the manuscript more robust. Some (minor) comments are below.

Lines 36-37: Add a reference to support this information.

Lines 77-80: Have the authors ensured that the isolates are clonally different? PFGE.

Line 423: What method was used to assess the quantity and quality of the extracted DNA?

Translated with DeepL.com (free version)

Author Response

REVIEWER 2:

Manuscript has been revised and modified, according to kind comments and suggestions of the reviewer. Changes are yellow marked in the manuscript:

Lines 36-37: Add a reference to support this information.

A Reference has been addded:

“[3] Skandalis, N.; Maeusli, M.; Papafotis, D.; Miller,S.; Lee, B.; Theologidis, I.; Luna, B. Environmental spread of antibiotic resistance. Antibiotics, 2021, 10, 640. https://doi.org/10.3390/antibiotics10060640

Lines 77-80: Have the authors ensured that the isolates are clonally different? PFGE.

We did not apply any molecular fingerprinting method to the isolates because our primary goal was to conduct a screening and survey of the evolution of resistance rates over time, using E. coli as an indicator of trends. While genotyping the isolates would have provided valuable insights into the distribution of E. coli clonal populations on the farm, this was not our main objective.

To address the reviewer's concerns, we have replaced the term “strain” with “isolates” in the manuscript we have discussed this point (Lines 271-274):

Our methodological approach does not allow to identify the specific pathways for which AMR E. coli reach and spread in the studied farm. More specific studies, tracking different clonal E. coli populations over time, should be undertaken to achieve this goal.”

We have also included a text about the usefulness of using E. coli as an indicator of environmental resistance (Lines 48-54):

E. coli has been proposed as the main indicator of the presence of antibiotic resistant bacteria and clinically relevant resistance-genes [4,11], due to its presence in animal and human gut, clinical relevance and ability to acquire conjugative plasmids. Moreover, commensal strains of E. coli reflect population exposure to antimicrobial selection pressure and thus can provide continuous evidence on trends [12].”

Line 423: What method was used to assess the quantity and quality of the extracted DNA?

A brief explanation about the method used to assess quantity and quality of extracted DNA, with the corresponding reference, has been added (Lines 501-504):

“Quantity (A260–A320) and quality (A260/A280 ratio > 1.7) of DNA was assessed by spectrophotometry (Qubit™ 4 Fluorometer, Invitrogen, Walthman, MA, USA) [67]”

Reviewer 3 Report

Comments and Suggestions for Authors

This study on the prevalence of antibiotic-resistant Escherichia coli in the poultry sector provides valuable insights into the spread of resistance genes through the food chain and environment. It addresses a critical public health issue by ensuring a comprehensive assessment of resistance levels at different stages of the rearing period through the sampling of one-day-old breeders, fecal samples, and boot swabs. Observing the decrease in resistance levels, including cephalosporin resistance and multiresistance, during rearing is valuable for understanding the dynamics of resistance development and persistence.

Abstract: The results presented in the abstract need to be clearly articulated for better readability. It would be significant to state the prevalence of different resistant genes at various stages. Adding information about the sampling methods used for collecting and processing samples is recommended. A brief statement should also be included about detecting resistance genes and determining susceptibility patterns.

Introduction: The introduction provides important information but could benefit from additional background information and a stronger argument for the study. Starting with a broader context about antimicrobial resistance (AMR) globally, mentioning it as a rising global health threat, and providing some statistics on morbidity, mortality, and economic burden would be beneficial. Including more relevant references would provide a comprehensive background. Consider identifying gaps in current knowledge that your study aims to fill.

 Methods: The methodology outlined is robust and comprehensive, with well-defined protocols and standard procedures. Sampling 14,500 breeding hens provides a robust dataset for statistical analysis. The use of one farm for sampling can be appropriate if the farm is representative of typical conditions in the poultry sector. However, this should be clearly stated, and any limitations associated with sampling from only one farm should be acknowledged. The methodology does not account for other environmental factors (e.g., feed, water quality, housing conditions) that could influence the results. The study focuses on specific antibiotic resistance genes (ARGs), potentially missing other relevant resistance genes, which can be discussed in the discussion section.

 Results and discussion: This section clearly describes the study's outcome. However, including more tables and graphs instead of paragraphs may help the reader understand the findings better. Separating the results from the discussion would improve clarity and help the reader. The lengthy combined results and discussion section is tiring for the reader, and using graphs would enhance the readability of the results. Their many vague sentences and repetitive content, which I recommend condensing for better clarity. Please refer to the manuscript comments for specific remarks.

 References: A few DOI links are incomplete (carefully check all)

Author Response

REVIEWER 3:

Manuscript has been revised and modified, according to kind comments and suggestions of the reviewer. Changes are yellow marked in the draft:

 ABSTRACT:

The results presented in the abstract need to be clearly articulated for better readability. It would be significant to state the prevalence of different resistant genes at various stages. Adding information about the sampling methods used for collecting and processing samples is recommended. A brief statement should also be included about detecting resistance genes and determining susceptibility patterns.

The abstract has been modified according to the reviewers' suggestions. However, as there is a limit to the length of the abstract (200 words maximum), we have had to limit the information provided.

 INTRODUCTION:

The introduction provides important information but could benefit from additional background information and a stronger argument for the study. Including more relevant references would provide a comprehensive background

We have added more background information and references, throughout the “Introduction” and “Discussion” sections

We have also better argued our study (Lines 91-105):

“There is strong evidence that the use of antibiotics in poultry production has led to the development of high resistance levels in the microbiome of animals, increasing the risk of transfer of these resistances from poultry-associated bacteria to human pathogens, with potentially serious consequences for public health [27]. Moreover, poultry farms have been shown to be a source of resistance spread to the environment [28]. Despite the restrictions on antimicrobial use in many countries, there have been multiple reports of antibiotic-resistant bacteria associated with poultry, which present food safety concerns [29]. At present, there is a growing interest in knowing the evolution of antibiotic resistance, not only in broilers, but throughout the entire production pyramid [30-33], to implement preventive measures at an early stage. Thus, in this study we aimed to determine the prevalence of antibiotic resistance in E. coli isolates obtained from a commercial breeder hen farm throughout the different stages of production (from day-old breeders to adult hens), as well as to detect cephalosporin and quinolone resistance genes in those E. coli isolates that showed resistance to cephalosporins and/or ciprofloxacin.”

Starting with a broader context about antimicrobial resistance (AMR) globally, mentioning it as a rising global health threat, and providing some statistics on morbidity, mortality, and economic burden would be beneficial.

According to reviewer suggestion, a new paragraph has been added (Lines 32-36):

“Antimicrobial resistance (AMR) is a rising global health threat. It is estimated that bacterial AMR was directly responsible for 1.27 million global deaths in 2019 and contributed to 4.95 million deaths. In addition, the World Bank estimates that AMR could result in US$ 1 trillion to US$ 3.4 trillion gross domestic product (GDP) losses per year by 2030 [1].

Consider identifying gaps in current knowledge that your study aims to fill.

We have highlighted this point (Lines 98-100):

“At present, there is a growing interest in knowing the evolution of antibiotic resistance, not only in broilers, but throughout the entire production pyramid [31-34], to implement preventive measures at an early stage.”

METHODS:

The methodology outlined is robust and comprehensive, with well-defined protocols and standard procedures. Sampling 14,500 breeding hens provides a robust dataset for statistical analysis. The use of one farm for sampling can be appropriate if the farm is representative of typical conditions in the poultry sector. However, this should be clearly stated, and any limitations associated with sampling from only one farm should be acknowledged. The methodology does not account for other environmental factors (e.g., feed, water quality, housing conditions) that could influence the results.

The study focuses on specific antibiotic resistance genes (ARGs), potentially missing other relevant resistance genes, which can be discussed in the discussion section.

According to reviewer’s comment, we have added a new paragraph discussing these two points (Lines 421-426):

“However, some limitations of the study must be considered. Although the sampled farm represents typical conditions in our poultry sector, resistance levels may vary among farms due to different rearing methodologies or environmental conditions (e.g., feed, water quality, housing conditions) that could influence the results [40,41,49,]. Moreover, this study focuses on specific ARG, potentially missing other relevant resistance genes [65].”

RESULTS AND DISCUSSION:

This section clearly describes the study's outcome. However, including more tables and graphs instead of paragraphs may help the reader understand the findings better.

Two graphs have been added (Lines 158 and 193)

Separating the results from the discussion would improve clarity and help the reader. The lengthy combined results and discussion section is tiring for the reader, and using graphs would enhance the readability of the results.

We have separated the “Results” and “Discussion” sections and adjusted the text to align with this new structure in the draft.

Their many vague sentences and repetitive content, which I recommend condensing for better clarity.

Manuscript has been carefully revised and the text has been modified to avoid vague sentences and repetitive content

REFERENCES: A few DOI links are incomplete (carefully check all)

All the references have been revised and completed.

Please refer to the manuscript comments for specific remarks.

MANUSCRIPT COMMENTS:

Line 47-49: add a citation

According to reviewers’ comments, we have added new references as well as more information (Lines 48-57):

“Escherichia coli (E. coli) is a natural inhabitant of the gut in poultry and one of the main microorganisms responsible for the spread of resistant genes among the environment, animals, and humans [9-10]. E. coli has been proposed as the main indicator of the presence of antibiotic resistant bacteria and clinically relevant resistance-genes [4,11], due to its presence in animal and human gut, clinical relevance and ability to acquire conjugative plasmids. Moreover, commensal strains of E. coli reflect population exposure to antimicrobial selection pressure and thus can provide continuous evidence on trends [12]. E. coli is especially useful to monitor extended-spectrum β-lactamases (ESBL) producing bacteria, because ESBL-producing Enterobacteraceae are the major cause of resistance to expanded-spectrum β-lactam antibiotics [4, 13].

Line 51: add citations

According to reviewers’ comments, we have changed redaction of this paragraph and added new references (Lines 58-62):

“Beta-lactams account for 60% of all antibiotics used worldwide, and are one of the most widely prescribed antibiotic classes [14]. The most frequent form of resistance to β-lactam antibiotics is the production of β-lactamases, mainly extended-spectrum β-lactamases (ESBL) and plasmidic AmpC (pAmpC) β-lactamases, which can hydrolyze penicillin and cephalosporins [15].”

Line 54: citations

A new citation has been added (Line 62):

[15]: Fisher, J. F.; Meroueh, S. O.; Mobashery, S. Bacterial resistance to beta-lactam antibiotics: compelling opportunism, compelling opportunity. Chem. Rev., 2005, 105, 395-424. https://doi.org/10.1021/cr030102i

Line 62-66: include more recent references

More recent references have been included:

  1. Aworh, M.K.; Kwaga, J.K.P.; Hendriksen, R.S.; Okolocha, E.C.; Harrell. E.; Thakur, S.. Quinolone-resistant Escherichia coli at the interface between humans, poultry and their shared environment- a potential public health risk. One Health Outlook, 2023, 5, 2. https://doi.org/10.1186/s42522-023-00079-0
  2. Truswell, A.; Lee, Z.Z.; Stegger, M.; Blinco, J.; Abraham, R.; Jordan, D.; Milotic, M.; Hewson, K.; Pang, S.; Abraham, S. Augmented surveillance of antimicrobial resistance with high-throughput robotics detects transnational flow of fluoroquinolone-resistant Escherichia coli strain into poultry. J. Antimicrob. Chemother., 2023, 78, 2878-2885. https://doi.org/10.1093/jac/dkad323”

Line 87: Recommending to include results stated in this section in a table for easy reference and also to increase the redability

All the results stated in Section 2.1. are included in Table 1. We have added a new sentence for easy reference (LINE 111 ):

“Table 1 and Figure 1 show the results of antibiotic resistance levels.”

 Line 107: this observation is from a one breeding farm, the authors could have discussed this observation referring the treatment protocols of this farm and the usage pattern of antibiotics.  though they have informed the two pathways of getting, it is not clear the vertical transmission of AMR. It is not clear that the authors are mentioning the bacterial infection pathways or the AMR gene transfer. Thus, presenting results clearly, help the reader to understand the reasoning given in the discussion.

This paragraph has been included in Discussion section, to clarify the understanding of the reasoning. According to reviewer’s comment, we have highlighted that vertical transmission is not clear, although it is supported by some other authors, providing a reference (Lines 260-264):

“Several studies [33,34] have shown the transmission of AMR throughout the hen production system, which may be due to two factors: a) a possible vertical transmission from parents to the offspring, caused by a possible infection in the hen’s uterus during egg formation, or by fecal contamination in the cloaca during egg-laying [35]

We have also discussed our results according to the treatment protocols of this farm (Lines 344-346):

“..confirming a reduction in the resistance levels with increasing age for all antimicrobials except for AMP and NA which were higher in pullets and adults, respectively. As indicated in “Material and Methods” section, only two antibiotic treatments were given to the hens during our study. This restrictive protocol is aimed to prevent antibiotic pressure on the animals. The lack of selective pressure avoids AMR spreading among the bacteria population.”

Line 117: From those reasoning, what could be the possible reason for the presence of AMR resistant E-coli in this farm? related the reasoning to your observation for clarity

We have outlined the possible reasons for the presence of AMR resistant E. coli in this farm, although we cannot be specific about which ones influence our results, because we did not perform molecular fingerprinting of different E. coli populations over time. We have now clarified this point in the text (Lines 271-274):

“Our methodological approach does not allow us to identify the specific pathways for which AMR E. coli reach and spread in the studied farm. More specific studies, tracking different clonal E. coli populations over time, should be undertaken to achieve this goal.”

Line 131: these are too vague statements, and the authors should consider specific reasoning for your results getting the supportive evidence from the literature

This paragraph has been rewritten according to reviewer’s comments, citing supporting evidence from other authors (Lines 275-279):

“Because of their importance in human medicine, the high observed rates of resistance and intermediate susceptibility to 3GC we have found in one-day-old hens are remarkable [18]. Similar to us, other authors have also detected high resistance levels to these antibiotics in E. coli isolated from both, one-day-old and one-week chicks, thus suggesting that 3GC resistance is present in hens since their first moments of life and [31,32,35].”

Line 161: the results obtained from this study is simply comparable, except CIP and NA, AMP, TE. However, the reasoning for the observed differences between studies, are not strong enough as the authors just mentioned that difference is breeding hens and broilers. 

We have added some more information to improve our discussion about the observed differences (Lines 308-315):

“These results are not surprising. EFSA report hisghlihts that resistance levels greatly differed among reporting countries and antimicrobials, showing important variations between and within food-producing animal populations and countries [4], due mainly to the absence of harmonized approaches using the same methodology among countries for survey studies [40]. It should also be considered that our study focused on breeding hens while EFSA reported data obtained from broilers and that rearing process can differ among farms, thus affecting the AMR spread and levels ([41,42].”

Line 176: I recommend concise writing separating the results and giving proper reasons for any observed difference between studies. Authors are listing the outcomes of the previous research and just mentioning the observed results of this study is different, but no proper reasoning and specific discussions.

“Results” and “Discussion” sections are now separated, and Discussion has been modified to improve reasoning of results, according to reviewer’s recommendations

Line 240: is this resistance levels or % resistance.

These are resistance levels, determined by the % of resistant strains, according to EFSA and CDC [4].

Line 244: again, the same comment, showing the resistant spectra in a grap and discussion of the observed pattern may help the reader to better understand

A graph has been added and discussed in “Discussion” section

Line 314: the other possble reason is the use of antibiotics at the early stages and the reduction of use at later stages, whcih allow the susseptible strain to be persistent

This argument pointed by the reviewer has been added to the text (Lines 378-379):

“Other possible reason is the use of antibiotics at the early stages and the reduction of use at later stages, which allow the susceptible strains to persist.”

Line 338: I suggest, rewarding "hense likely to introduce the resistant strains.....

This sentence has been changed according to reviewer suggestion (Line 366)

Line 397: delete. “colonies” has been deleted (Line 454)

Round 2

Reviewer 1 Report

Comments and Suggestions for Authors

Authors have incorporated comments and suggestions. The manuscript is now improved and is acceptable for publication 

Author Response

The authors are grateful for your review